# Anti SARS-CoV-2 Monoclonal Antibodies in Pre-Exposure or Post-Exposure in No- or Weak Responder to Vaccine Kidney Transplant Recipients: Is One Strategy Better than Another?

**DOI:** 10.3390/v16030381

**Published:** 2024-02-29

**Authors:** Anais Romero, Charlotte Laurent, Ludivine Lebourg, Veronique Lemée, Mélanie Hanoy, Frank Le Roy, Steven Grange, Mathilde Lemoine, Dominique Guerrot, Dominique Bertrand

**Affiliations:** 1Department of Nephrology and Hemodialysis, Hôpital de la Croix Rouge, 76230 Bois Guillaume, France; anais.romero@croix-rouge.fr; 2Department of Nephrology, Transplantation and Hemodialysis, 1 Rue de Germont, Rouen University Hospital, 76000 Rouen, France; charlotte.laurent@chu-rouen.fr (C.L.); ludivine.lebourg@chu-rouen.fr (L.L.); melanie.hanoy@chu-rouen.fr (M.H.); frank.le-roy@chu-rouen.fr (F.L.R.); steven.grange@chu-rouen.fr (S.G.); mathilde.lemoine@chu-rouen.fr (M.L.); dominique.guerrot@chu-rouen.fr (D.G.); 3Department of Virology, Rouen University Hospital, 76000 Rouen, France; veronique.lemee@chu-rouen.fr; 4INSERM U1096, University of Rouen Normandy, 76000 Rouen, France

**Keywords:** COVID-19, monoclonal antibodies, SARS-CoV-2, kidney transplantation

## Abstract

**Background:** Kidney transplant recipients (KTRs) are likely to develop severe COVID-19 and are less well-protected by vaccines than immunocompetent subjects. Thus, the use of neutralizing anti–SARS-CoV-2 monoclonal antibodies (mAbs) to confer a passive immunity appears attractive in KTRs. **Methods:** This retrospective monocentric cohort study was conducted between 1 January 2022 and 30 September 2022. All KTRs with a weak antibody response one month after three doses of mRNA vaccine (anti spike IgG < 264 (BAU/mL)) have received tixagevimab-cilgavimab in pre-exposure (group 1), post-exposure (group 2) or no specific treatment (group 3). We compared COVID-19 symptomatic hospitalizations, including intensive care unit hospitalizations, oxygen therapy, and death, between the three groups. **Results:** A total of 418 KTRs had SARS-CoV-2 infection in 2022. During the study period, we included 112 KTRs in group 1, 40 KTRs in group 2, and 27 KTRs in group 3. The occurrence of intensive care unit hospitalization, oxygen therapy, and COVID-19 death was significantly increased in group 3 compared to group 1 or 2. In group 3, 5 KTRs (18.5%) were admitted to the intensive care unit, 7 KTRs (25.9%) needed oxygen therapy, and 3 KTRs (11.1%) died. Patients who received tixagevimab-cilgavimab pre- or post-exposure had similar outcomes. **Conclusions:** This retrospective real-life study supports the relative effectiveness of tixagevimab-cilgavimab on COVID-19 infection caused by Omicron, used as a pre- or post-exposure therapy. The continued evolution of Omicron variants has made tixagevimab-cilgavimab ineffective and reinforces the need for new therapeutic monoclonal antibodies for COVID-19 active on new variants.

## 1. Introduction

Severe acute respiratory syndrome coronavirus 2 (SARS-CoV-2) infection caused a global pandemic that affected France in March 2020. Over 3 years, more than 160,000 people died [1], particularly in immunocompromised persons like kidney transplant recipients (KTRs), for whom the mortality rate was close to 22% at the start of the pandemic [2].

One response to this pandemic was vaccination, which was rapidly recommended through international guidelines [3,4]. Despite the rapid implementation of a third dose of mRNA vaccine in some countries, both humoral and cellular responses to vaccine against SARS-CoV-2 are reduced in KTRs [5], resulting in increased incidence of severe infection and mortality, including in fully vaccinated patients [6]. Moreover, the Omicron variant was identified in November 2021 in Botswana, South Africa, and quickly spread worldwide to become the predominant variant. Due to his immune escape profile, vaccination results in reduced neutralizing activity against Omicron compared with the ancestral strain [7].

In this context, monoclonal antibodies (mAbs) providing passive immunization have been developed to enhance immunity against SARS-CoV-2 in immunocompromised patients [8]. French health authorities granted authorization in December 2021 for tixagevimab-cilgavimab in pre-exposure prophylaxis for immunocompromised patients with a complete vaccine scheme and no or weak humoral response (<264 binding antibody units (BAU/mL)) one month after the last injection. It’s a combination of two fully human, SARS-CoV-2 neutralizing monoclonal antibodies, which are derived from antibodies isolated from B cells obtained by persons infected with SARS-CoV-2. PROVENT study assessed tixagevimab-cilgavimab for pre-exposure prophylaxis against symptomatic COVID-19. Relative risk reduction for symptomatic COVID-19 was 76.7% in the tixagevimab-cilgavimab group. Efficacy is estimated to last at least 6 months [9]. Tixagevimab-cilgavimab can also be used as an early treatment in high-risk patients developing moderate-to-severe COVID-19 [10]. Nevertheless, its effectiveness on Omicron variants is largely debated because of their immune escape to the vast majority of mAbs [11,12,13,14,15,16].

Although data are available on the efficacy of tixagevimab-cilgavimab in immunocompromised patients [17,18], an assessment of real-world efficacy in solid organ transplant (SOT) recipients has been limited [12]. To date, no study has compared the pre- and post-exposure strategy of SARS-CoV-2 mAbs in non- or weak responders to vaccine KTRs during the Omicron period. We report here the impact of these two strategies on the incidence of symptomatic COVID-19 and COVID-19-related hospitalizations, including intensive care unit hospitalizations and death in a cohort of KTRs during 2022.

## 2. Materials and Methods

This retrospective cohort study was conducted in an adult kidney transplant unit of one French University Hospital (Rouen) between, 1 January 2022 and 30 September 2022. According to French law (loi Jardé), because this study was anonymous and retrospective, institutional review board approval was not required.

We retrospectively identified all KTRs infected with SARS-CoV-2 during 2022. Demographic data, comorbidities, history of previous COVID-19, anti-SARS-CoV-2 vaccination, mAbs injections, kidney transplant data, immunosuppressive therapy, details on COVID-19 characteristics, management, and clinical outcomes were collected. Acute kidney injury was defined as an increase in serum creatinine of >50% [19].

All adult KTRs considered as low- or non-responders to the vaccine (anti SARS-CoV-2 spike IgG < 264 BAU/mL 1 month after 3 injections of mRNA vaccine) with a diagnosis of proven COVID-19 were included in the study. Diagnosis of COVID-19 was based on the PCR (Polymerase Chain Reaction) test carried out on nasopharyngeal swabs. Genome sequencing from PCR was performed when suitable. Patients who received double solid organ transplantation were also deemed eligible. All patients received mRNA (BNT162b2 vaccine or mRNA-1273). Patients could have received pre-exposure prophylaxis with casiriviab-imdevimab prior to the use of tixagevimab-cilgavimab. Excluded patients are summarized in Figure 1. KTRs with a SARS-CoV-2 infection after September 30th were excluded because the BQ.1.1 variant was predominant, and tixagevimab-cilgavimab was considered ineffective [20]. KTRs who received both pre- and post-exposure tixagevimab-cilgavimab were also excluded. A database was present in the hospital to identify episodes of COVID-19 in KTRs, and all patients with a history of previous COVID-19 were excluded. In addition, KTRs having received another curative treatment, such as sotrovimab, were excluded.

According to local treatment protocol and physician decision, treatment availability, and circulating variants, all immunocompromised patients with a weak antibody response one month after three doses of mRNA vaccine (anti spike IgG < 264 (BAU/mL)) could have received:
−Tixagevimab-cilgavimab in pre-exposure COVID-19 according to this scheme: intramuscular injections of 150 mg tixagevimab-150 mg cilgavimab between 23 December 2021 and February 2022, then 2 intramuscular injections of 150 mg tixagevimab-150 mg cilgavimab between April 2022 and May 2022 and then 2 intramuscular injections of 300 mg tixagevimab-300 mg cilgavimab between November and December during consultation. The last prophylaxis injection must have been less than 6 months old. (Group 1: prophylactic group)−Tixagevimab-cilgavimab curative treatment: mAbs should be administered within 5 days of onset of symptoms in KTRs without oxygen therapy, independently of the presence of symptoms or the reason for testing (symptoms, COVID-19 contact or systematic testing). Tixagevimab-cilgavimab in curative treatment was given intravenously at 600 mg a day in the hospital. (Group 2: curative group)−No specific treatment (Group 3: no treatment). The reasons why the KTRs did not receive mAbs prophylaxis were mostly patient refusal or curative anticoagulant.

The last follow-up was on 1 June 2023. Our evaluation criteria were COVID-19 symptomatic hospitalizations, including intensive care unit hospitalizations, oxygen therapy, and death.

Statistics were performed using Statview version 5.0 (SAS Institute Inc., Brie Comte Robert, France). Quantitative variables were expressed as median with their interquartile range (IQR) or mean with their standard derivation (SD). Categorical variables were described as absolute numbers and percentages. *T*-tests were used for the comparison of quantitative variables between groups and Pearson’s Chi-squared test for categorical variables.

## 3. Results

Between 1 January and 31 December 2022, 418 KTRs had COVID-19. We excluded 15 unvaccinated patients, 5 patients partially vaccinated, 29 patients with a past history of SARS-CoV-2 infection and 19 KTRs who received Sotrovimab as a curative treatment. Moreover, 75 KTRs were infected and, therefore, potentially related to BQ.1.1 and were excluded. Twelve KTRs received tixagevimab-cilgavimab pre- and post-exposure and were also excluded from the analysis. None was excluded for missing data.

A flow diagram describing the patient samples and exclusion is shown in Figure 1: 179 KTRs were included in the final analysis. One hundred and twelve KTRs received pre-exposure tixagevimab-cilgavimab before COVID-19 (group 1), 40 KTRs received post-exposure tixagevimab-cilgavimab (group 2), and 27 KTRS did not receive tixagevimab-cilgavimab (group 3). No severe adverse events were reported among treated patients.

The general characteristics of patients are summarized in Appendix A. The average age was 56.7 ± 14.1 years, and 104 (58%) were men. SARS-CoV-2 infection was identified after a median of 64.6 months (interquartile range: 33.5–146) from kidney transplantation.

Appendix A depicts COVID-19 presentation, management, and clinical outcomes in the entire cohort. One hundred and forty-four KTRs (80.4%) were symptomatic. The most frequent symptom on admission was cough, fever, rhinitis, and asthenia in 75 KTRs (44.1%), 50 KTRs (29.4%), 49 KTRs (28.8%), and 46 KTRs (27%), respectively. Viral genotypic data collected at illness were available for 32 KTRs (17.9%): 14 BA.1 between January and February, 10 BA.2 between March and May, and 8 BA.5 between June and September. Among the 179 infected KTRs, 32 KTRs (17.9%) were admitted to the hospital, including 7 KTRs (3.9%) in intensive care. Nasal oxygen therapy was administered in 21 KTRs (11.7%), and antibiotics were administered in 21 KTRs (11.7%). Fourteen KTRs (7.8%) received dexamethasone, and 2.8% received convalescent plasma. Acute kidney injury occurred in 23 KTRs (12.9%). The mortality rate was 3.3%.

Table 1 shows the characteristics of these groups. There was no significant difference between groups except for immunosuppressive therapy. Group 1 had more corticosteroids than group 2 (*p* = 0.03). Group 2 had less belatacept than group 3 (*p* = 0.04) and tacrolimus, though levels were higher in group 2 than group 3 (*p* = 0.04). eGFR was higher in group 2 than in group 3 (*p* = 0.05). Group 1 had received, on average, 1.3 tixagevimab-cilgavimab injection before infection. The time between the last injection and SARS-CoV-2 was a median of 2.2 months (interquartile range: 1–3.3). Curative treatment was conducted on a median of 2 days (interquartile range: 0.8–3.1) after the onset of symptoms.

Table 2 shows outcomes according to the group. The occurrence of intensive care unit hospitalization, oxygen therapy, and COVID-19 death was significantly increased in group 3 compared to group 1 or 2. Indeed, in group 3, 5 KTRs (18.5%) were admitted to intensive care, 7 KTRs (25.9%) needed oxygen therapy, and 3 KTRs (11.1%) died. Patients who received tixagevimab-cilgavimab pre- or post-exposure had similar outcomes.

## 4. Discussion

Our study suggests for the first time a comparable efficacy of pre- or post-exposure tixagevimab-cilgavimab in low- or non-responders to vaccine KTRs with a SARS-CoV-2 infection. We also reported that the use of specific monoclonal antibodies in non- or low-responders patients is associated with lower mortality.

Prophylaxis mAbs have been widely used, and their effectiveness is no longer demonstrated. Post-exposure administration of casirivimab-imdevimab prevented 84% of infections in a randomized clinical trial, which was conducted before Omicron circulation [21]. Levin et al. show in a randomized clinical trial a relative risk reduction of 76% for symptomatic COVID-19 in the pre-exposure tixagevimab-cilgavimab administration. In the 6-month follow-up, five patients with severe or critical COVID-19 were reported, all of which occurred in the placebo group. Nevertheless, only 3.2% of patients were on immunosuppressive therapy in this study [9]. In our previously published study, pre-exposure prophylaxis with tixagevimab-cilgavimab reduced COVID-19 infection in insufficiently immunized patients (6.8% versus 35%; *p* < 0.001) [12]. Kaminsky et al. reported that 77% of KTRs who did not respond to the vaccine received tixagevimab-cilgavimab as a preventive measure. Among them, 12.3% had symptomatic COVID-19 compared to 43.3% in patients who did not receive prophylaxis (HR: 0.011; 95% CI (0.063–0.198; *p* < 0.001)). The use of mAbs also reduced the mortality of KTRs. Two patients (1.8%) died in the prophylaxis group in our study [22]. This result is concordant with our recently published study, in which we reported no death among 28 KTRs who had received tixagevimab-cilgavimab prophylaxis during the Omicron wave [12]. A similar mortality rate was found in other studies [15,22]. We did not show any difference in the rate of symptomatic COVID-19 or hospitalization between KTRs who received or did not receive the mAbs requirement for intensive care hospitalization, and death was significantly higher in KTRs who did not receive mAbs. KTRs who received mAbs demonstrated the same efficacy as those who developed post-vaccination antibodies despite the Omicron variant [12]. The partial efficacy of mAbs on the occurrence of symptomatic COVID-19 in our study is possibly related to a low-dose treatment [23].

Post-exposure treatment also seems to be a valuable strategy for low- or non-responder KTRs with SARS-CoV-2 infection. Peak concentrations of tixagevimab-cilgavimab are reached 1 h after intravenous administration and are followed by a rapid onset of action [24]. Levin et al. show in a randomized clinical trial a relative risk reduction of 33% RT-PCR-positive symptomatic COVID-19 with post-exposure tixagevimab-cilgavimab versus placebo-treated participants [8]. In Gueguen’s study, a total of 80 KTRs received mAbs between February 2021 and June 2021. They were matched to 155 controls. Early infusion of mAbs reduces the risk of severe COVID-19 with a 10-fold lower risk of admission to the intensive care unit and death [25]. Gupta et al. show infection treatment with Sotrovimab achieved 85% efficacy in preventing COVID-19-related hospitalizations or deaths: 3 patients (1%) in the sotrovimab group as compared with 21 patients (7%) in the placebo group had disease progression leading to hospitalization or death (*p* = 0.002) [26]. In a retrospective study reported by Benotmane et al., post-exposure tixagevimab-cilgavimab in KTRs showed a decrease in COVID-19-related care hospitalizations (*p* < 0.01), oxygen need (*p* = 0.04), but no difference in deaths. These results were in line with our results and seemed even better, but inclusion criteria were not strict: 19% received tixagevimab-cilgavimab pre- and post-exposure, 4% were unvaccinated, 3.8% had a past history of previous COVID-19 [14] compared to our study in which we excluded these patients in order to limit multiple biases.

These results are discordant with those observed in vitro, which suggested that mAbs may have reduced efficacy against Omicron [11]. Omicron was completely or partially resistant to neutralization in vitro by all monoclonal antibodies tested. Patients who received tixagevimab-cilgavimib had a low level of neutralizing activity, and only 9.5% of them were able to neutralize the Omicron variant compared with 71% of patients who were infected with SARS-CoV-2 [13]. The World Health Organization designated Omicron as a variant of concern on 26 November 2021. Successive sub-lineages of Omicron have spread worldwide since the identification of BA.1 in November 2021. Over 80% of the population has been infected with either Omicron variant in less than a year [15]. However, the occurrence of the Omicron variant has been associated with reduced severity of COVID-19. Ulloa et al. matched patients with COVID-19 infection Delta/Omicron according to sex, age, and vaccination status. In comparison with Delta, Omicron caused fewer hospitalizations (1.4% versus 0.3%) and deaths (0.3% versus 0.03%). HR: 0.12 (95% CI, 0.04–0.37) [27]. Benotmane et al. reported, in a retrospective study without a control group, the occurrence of breakthrough COVID-19 cases despite prophylaxis with tixagevimab-cilgavimab in KTRs. Analyzed sera from 29 immunocompromised individuals up to 1 month after administration of tixagevimab-cilgavimib show that patients treated have high antibody levels, which efficiently neutralized the Delta variant. As compared to the Delta variant, neutralizing titers were more markedly decreased against BA.1 (344-fold) than BA.2 (9-fold). In our study, identification of the variant was only possible in 32 patients (17.7% of cases) [28]. However, it has been unclear whether reduced in vitro susceptibility translated to reduced clinical effectiveness—particularly for patient populations with chronic substantially immunocompromised states. We previously reported worse outcomes during BQ.1.1 period in KTRs who received tixagevimab-cilgavimab [29]. In our cohort of KTRs, seven patients presented a SARS-CoV-2 BQ.1.1 proven infection. Among them, all were hospitalized, including three KTRs in intensive care; two KTRs died. Jordan et al. showed, on the other hand, a 39% reduction in COVID-19 infections after tixagevimab-cilgavimib use during the period of Omicron. In analyses adjusting for demographic, clinical, and COVID-19 exposure factors, any tixagevimab-cilgavimib treatment was associated with lower infection risk (OR 0.52, 95% CI 0.27–0.96, *p* = 0.039) throughout the surveillance period including when more resistant BQ.1 and BQ.1.1 subvariants had emerged. High titer neutralizing antibodies to spike protein lasting for more than 30 weeks after administration were present [30]. Moreover, in January 2023, tixagevimab-cilgavimab was no longer authorized, given the concern for the lack of effective neutralization of the newly emergent Omicron subvariants at that time [31].

The question of prophylactic versus curative treatment will still be relevant when new effective mAbs are available. The preventive strategy is time-consuming but seems to limit the number of new cases. A curative strategy seems to be a good alternative, less difficult to implement, limiting the number of severe SARS-CoV-2 infections and deaths. Moreover, mAbs could drive the emergence of SARS-CoV-2 variants, so their use must be rationed. Indeed, patients treated with various mAbs might develop evasive Spike mutations with remarkable speed and high specificity to the targeted mAb-binding sites. Gupta et al. showed patients receiving bamlanivimab, bamlanivimab/etesevimab, or casirivimab/imdevimab mostly carried Alpha subvariants. Whereas all patients treated with sotrovimab carried Omicron subvariants, the most common being 21K/BA.1 with the S:R346K substitution [32].

Other alternative therapies exist, such as convalescent plasma, which is a transfusion of blood plasma from patients who have recovered from COVID-19. Libster et al. showed early administration of high-titer convalescent plasma against SARS-CoV-2 to mildly ill, infected older adults reduced the progression of COVID-19. (relative risk, 0.52; 95% confidence interval [CI], 0.29 to 0.94; *p* = 0.03) [33]. Current evidence shows that convalescent plasma does not improve survival or reduce the need for mechanical ventilation, while it has significant costs. It’s, therefore, no longer recommended for use [34]. The direct-acting small molecule SARS-CoV-2 antivirals have received approval or emergency use authorization. They do not target the variable spike protein but target the conserved viral RNA-dependent RNA polymerase or the conserved viral main protease. Among them is Remdesivir, a monophosphoramidate prodrug of the nucleoside GS-441524 [35]. Nirmetrelvir/ritonavir could also be an alternative to prevent severe infections as a post-exposure treatment. However, there are many pharmacological interactions with nirmetrelvir/ritonavir, and they are contraindicated if eGFR < 30 mL/min/1.73 m^2^ [36]. A benefit—risk assessment tailored to the individual patient should be considered to guide the choice of the most appropriate option.

Despite its potential bias due to the retrospective design, the strength of our study is the number of patients included. Strict inclusion and exclusion criteria lead to a more interpretable study. As an example, patients with a past history of COVID-19 were excluded because they demonstrated a higher immune response [37]. The duration of follow-up is relatively prolonged, allowing for a reliable estimation of morbidity and mortality from COVID-19. Few studies have examined the use of specific mAbs between January and September 2022; the majority stopped in spring [12,13,15]. One major limitation of our study is the use of quantitative anti-spike antibodies for the evaluation of vaccine responses. The standard is the measurement of neutralizing antibodies against particular variants and subvariants, with additional recognition of the importance of T-cell responses. Moreover, the three groups (pre-exposure prophylaxis, treatment, and neither) were assigned per clinician choice and likely were not comparable at baseline concerning immunosuppressive treatment. Therefore, assessments of the relative efficacy of these interventions cannot be reliably ascertained.

This retrospective real-life study supports the relative effectiveness of tixagevimab-cilgavimab on COVID-19 infection caused by Omicron, used as a pre- or post-exposure therapy. The continued evolution of Omicron variants reinforces the need for new therapeutic monoclonal antibodies for COVID-19 active on new variants of Omicron, such as BQ.1.1 and XBB variants.

## Figures and Tables

**Figure 1 viruses-16-00381-f001:**
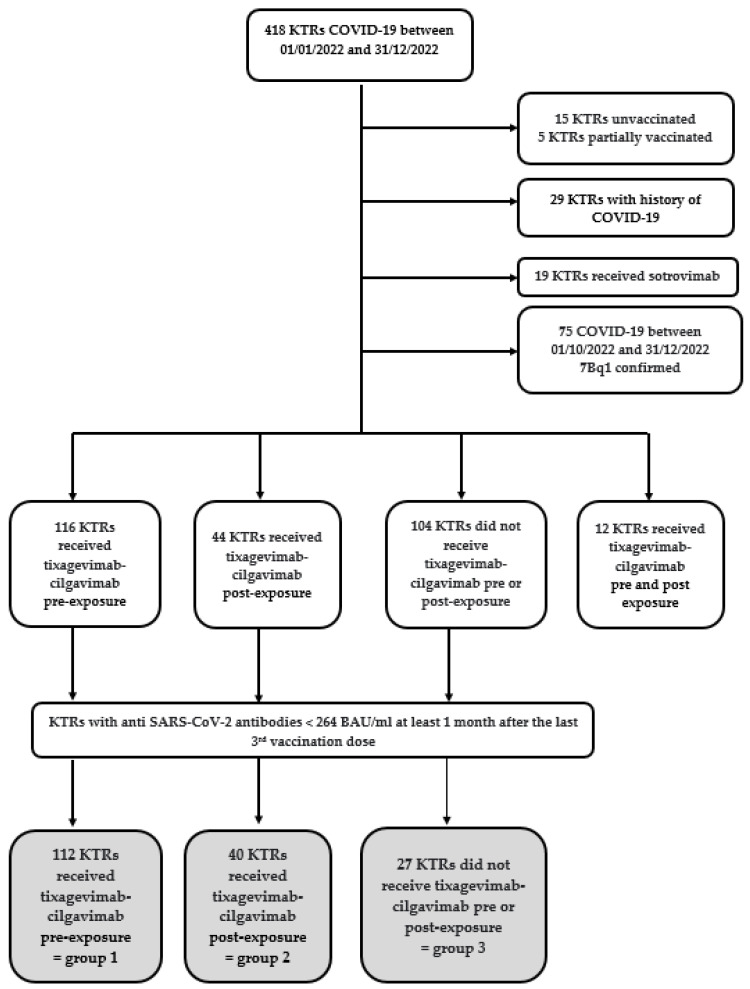
Flowchart. BAU, binding antibody unit; COVID, coronavirus; KTRs, kidney transplant recipients; SARS-CoV-2, severe acute respiratory coronavirus-2.

**Table 1 viruses-16-00381-t001:** Characteristics of KT in the three groups.

Characteristics	Group 1 n = 112	p Group 1 vs. 2	Groupe 2 n = 40	p Group 2 vs. 3	Group 3 n = 27	p Group 1 vs. 3
Age, years	57.2 ± 14.2	0.49	55.8 ± 14.5	0.55	56.8 ± 13.5	0.89
Male sexe, n (%)	66 (58.9)	0.84	23 (57.5)	0.87	15 (55.5)	0.72
Time from KT, median (range)	63.6 (35–125)	0.32	88.2 (31–166)	0.79	63.4 (32–141)	0.74
First transplantation, n (%)	93 (83)	0.29	35 (87.5)	0.33	24 (88.9)	0.07
**Induction therapy for KT n (%):**		0.78		0.32		0.23
ATG	51 (46)		18 (47)		14 (54)	
Anti-R-IL2	60 (54)		20 (53)		12 (46)	
eGFR, mL/min per 1.73 m^2^	46.1 ± 21.2	0.47	43.5 ± 18.6	0.3	38.5 ± 18.3	0.05
Body mass index, kg/m^2^	27.2 ± 5.1	0.64	27.0 ± 5.1	0.71	26.2 ± 4.5	0.31
**Nephropathy:**		0.11		0.35		0.54
Diabetes, n (%)	6 (5.4)		1 (2.5)		1 (3.7)	
Glomerulonephritis, n (%)	48 (42.8)		14 (35)		12 (44.4)	
Unkown cause, n (%)	6 (5.4)		6 (15)		2 (7.4)	
Interstitial nephropathy, n (%)	10 (8.9)		2 (5.0)		2 (7.4)	
Polycystic kideney, n (%)	22 (19.6)		10 (25)		4 (14.8)	
Malformative uropathy, n (%)	11 (9.8)		6 (15)		3 (11.1)	
Hypertension, n (%)	8 (7.1)		1 (2.5)		3 (11.1)	
Other cause, n (%)	1 (0.9)		0 (0)		0 (0)	
**Comorbidities:**						
Hypertension, n (%)	89 (79.5)	0.46	34 (85)	0.98	23 (85.2)	0.51
Diabetes mellitus, n (%)	32 (28.6)	0.41	9 (22.5)	0.69	5 (18.5)	0.26
Cardiac disease, n (%)	39 (34.8)	0.54	12 (30)	0.23	12 (44.4)	0.38
Vascular disease, n (%)	11 (9.8%)	0.57	3 (7.5)	0.99	2 (7.4)	0.62
Respiratory disease, n (%)	13 (11.6%)	0.33	7 (17.5)	0.63	6 (22.2)	0.14
Cancer, n (%)	22 (19.6%)	0.78	7 (17.5)	0.63	6 (22.2)	0.75
**Immunosupressive drugs at inclusion:**						
Mycophenolic acid, n (%)	97 (86.6)	0.91	35 (87.5)	0.49	22 (81.5)	0.48
Azathioprine, n (%)	6 (5.4)	0.61	3 (7.5)	0.61	3 (11.1)	0.27
Ciclosporin, n (%)	7 (6.2)	0.09	6 (15)	0.65	3 (11.1)	0.37
mTOR inhibitor, n (%)	3 (2.6)	0.3	0 (0)	1	0 (0)	0.39
Tacrolimus, n (%)	84 (75)	0.73	29 (72.5)	0.15	15 (55.5)	0.04
Belatacept, n (%)	17 (15.2)	0.43	4 (10)	0.04	8 (29.6)	0.07
Corticosteroids, n (%)	70 (62.5)	0.03	17 (42.5)	0.45	14 (51.8)	0.29

Data are given as mean ± SD, median ± IQR, mTOR, mammalian target of rapamycin; KT, kidney transplantation; ATG, antithymoglobuline; R-IL2, IL-2 receptor; n, number.

**Table 2 viruses-16-00381-t002:** Outcomes of KT according to the group.

	Group 1 n = 112	p Group 1 vs. 2	Group 2 n = 40	p Group 2 vs. 3	Group 3 n = 27	p Group 1 vs. 3
Symptomatic COVID-19, n (%)	90 (80.3)	0.31	35 (87.5)	0.08	19 (70.3)	0.25
COVID-19-related hospitalization, n (%)	17 (15.1)	0.71	7 (17.5)	0.24	8 (29.6)	0.07
COVID-19 related hospitalization in intensive care unit, n (%)	2 (1.8)	0.4	0 (0)	0.005	5 (18.5)	0.0003
Oxygen therapy, n (%)	12 (10.7)	0.29	2 (5)	0.01	7 (25.9)	0.04
COVID-19-related death, n (%)	2 (1.8)	0.77	1 (2.5)	0.14	3 (11.1)	0.02

n, number; COVID-19, coronavirus disease 19.

## Data Availability

The data presented in this study are available on request from the corresponding author.

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
