# Peer review of "Anti SARS-CoV-2 Monoclonal Antibodies in Pre-Exposure or Post-Exposure in No- or Weak Responder to Vaccine Kidney Transplant Recipients: Is One Strategy Better than Another?"

_viruses, 2024, doi:10.3390/v16030381_

Round 1
Reviewer 1 Report
Comments and Suggestions for Authors
Congratulations to the authors for the work done. I understand that although the subject and the results obtained may not be novel, the study may be of interest in order to present different strategies to address SARS-COVID19 infection in immunocompromised individuals.
As for the paper, just a couple of comments in order to improve the manuscript:
In the methods section: the inclusion and exclusion criteria should be more clearly explained.
And if, as the authors comment, patients who have covid from September onwards are excluded, the study period should be from January to September and not from January to December as mentioned.
The possible biases of the study should also be explained in this section.
Author Response
Thank you for your review.
We adressed all the comments in the revised mansucript.

Reviewer 2 Report
Comments and Suggestions for Authors
This paper by Romero et al reports on the use of tixagevimab-cilgavimab for prevention and treatment of COVID 19 infection in the era of Omicron variants in kidney transplant patients at risk for COVID 19 infection due to low immune responses to vaccines. This study examined the efficacy of tixagevimab-cilgavimab (evusheld) in 3 patient groups, those who received treatment prior to COVID 19 infections, those who were given evusheld after infection and those who did not recieve evusheld. Overall, the authors found that there was a significant protective effect in prevention of serious complications of infection in the two groups that received evusheld compared with those not treated. This included a prevention of hospitalization and deaths. Critique: As the authors point out, the efficacy of evusheld in prevention of infectons andserious complications of COVID 19 infections has been demonstrated previously, thus, the findings of this investigation are not unexpected. However, it is of interest that this protection did extend into the era of Omicron dominant infections.This was also reported in a recent paper (Jordan S et al Transplant Infect. Dis. October 2023). This paper should be referenced. Those investigators showed a 39% reduction in COVID 19 infections after evusheld use during the period of Omicron dominant infections. I think the authors would improve their manuscript by focusing on this point and why it differs from data obtained by in vitro assay neutralization assays that the FDA used to cancel evusheld in US in 12/22. I would also like the authors to focuse on other issues in the discussion, namely the immunosuppressive protocols and how this influenced immune responses to vaccine and if this was altered prior to vaccination.In addition, did the authors look at T-cell immunity to the vaccines which may differ from antibody responses. I think the discussion is too long and could be condensed. The authors need not discuss the data supporting the use of monoclonals in COVID 19 infection, this has been well established.Second, the authors mention that the use of monoclonals could increase resistant variants. I would like for them to give evidence for this. Other possible alternatives to the monoclonals which are no longer available should also be discussed. Here, the authors could focus on IVIg which contains high titers of IgG to ancestral and Omicron variants as well as anti-nucleocapsid antibodies. Finally, attention to English syntax is important, many errors noted.,I also noted that various font sizes were used in the manuscript. This should be corrected.
Comments on the Quality of English Languageneeds to be improved. Words like cured and trimester are not common English syntax in the situations discussed in the paper
Author Response

(The authors gave the same response as above.)
